# Between Rocks and 'High Places': On Religious Architecture in the Iron Age Southern Levant

Diederik J. H. Halbertsma and Bruce Routledge *

Department of Archaeology, Classics and Egyptology, University of Liverpool, Liverpool L69 3BX, UK; diederik@liverpool.ac.uk
* Correspondence: bruce.routledge@liverpool.ac.uk

**Abstract:** In this paper we examine why common methodologies for determining 'religious architecture' do not account for the diverse and fluid ways in which religious behavior can be expressed. We focus on religious architecture from the Iron Age Southern Levant highlighting certain sites that 'fall through the cracks' of current taxonomies. We propose a different way of approaching evidence for religious practice in the archaeological record, viewing religion as one dimension of social action made visible along a spectrum of ritualization.

**Keywords:** religion; architecture; Iron Age; Southern Levant

## 1. Introduction

Religion, ritual and architecture form a problematic nexus for archaeologists. On the one hand, it is difficult to deny the affective power of architecture as spaces that afford and constrain human bodies in terms of movement and sensorial experience (Kraftl and Adey 2008). Since somatic, sensorial and emotive affect is also a quality of ritual practice (Meyer 2016), it follows that architecture and ritual can be intimately linked as components of the embodied experience of religion (Verkaaik 2013). On the other hand, to identify architecture as religious in an archaeological context is to invoke a series of well-known interpretive problems. Are there universal attributes of religious architecture? Indeed, how does one determine the specifically religious affect generated by architecture if religion itself cannot be treated as an essential and distinct social domain (Asad 1993)?

One could argue that the ancient Near East offers us an emic solution to these interpretive problems. Afterall, there are an abundance of textual representations of temples and temple building from across the region and over several millennia (Boda and Novotny 2010). This textual record shows the emic conceptualization of the temple as a distinct institution that acted as the house and estate of a deity. This pairs well with an archaeological record marked by sequences of temples that can often be traced back through centuries or even millennia at the same locale, such as the consecutive temples at Eridu (Safar et al. 1981, p. 111), Aleppo (Kohlmeyer 2009), or Megiddo (Susnow 2020, p. 149). This is also reflected in the formal regularity of many temples such that typologies of temple architecture can been formed for different periods and regions of the ancient Near East (e.g., Heinrich 1982; Matthiae 1975). Hence, one need not posit religion as a discreet domain in the Ancient Near East to recognize temples as focal places for regularized religious activities. Such spatio-temporal regularity is a key component of religious experience as Janico Albrecht and colleagues have noted:

> "People perform religious communication in specific spatial and temporal contexts and in doing so produce specifically religious space and time: for instance, processions delimit territory, a complex ritual sequence transforms a day into a festival. Conversely, a temple presents itself as the preferred place to contact the divine, a holiday demands gifts for the gods.". (Albrecht et al. 2018, p. 572)

However, treating the ancient Near Eastern temple as an emic solution to the problem of religious architecture introduces new problems. Not infrequently, archaeologists working in the Near East encounter architectural settings for ritual activity that are more ambiguous than the classic temples of our typologies. These are contexts in which distinctions between "sacred" and "profane" space are problematic to draw (cf. Aldenderfer 2012, p. 23). Ironically, many such examples are found in the Iron Age of the Southern Levant.

We say 'ironically' because the Iron Age of the Southern Levant is the time and place central to Biblical Archaeology, and few things are more biblical than the idea of a national temple built in a sacred capital. Unfortunately, unlike Mesopotamia and Egypt, the Iron Age of the Southern Levant lacks the textual and iconographic evidence that would allow us to identify temples as discrete institutions dedicated to named deities with personnel, property, ritual calendars, etc. Certainly, there are candidates for temples, such as at Arad (Herzog 2002), Khirbet Ataruz (Ji 2019), Tel Dan (Arie 2008, pp. 7–11), Tel Moza (Kisilevitz 2015), Tell Qasile (Mazar 1980), Nahal Patish (Nahshoni and Ziffer 2009) and Tell eṣ-Ṣafi/Gath (Dagan et al. 2018). However, none of these buildings share consistent architectural features and are classified as temples due to their size, features and finds. Complex 650 at Tel Miqne/Ekron might be considered an exception in that it is identified as the house built for "*Pt[.]yh*" in an inscription found within the building (Gitin 2012). However, despite efforts to find parallels in Assyria and Cyprus (Gitin 2012) or Northern Syria (Nishiyama 2012), the overall uniqueness of this structure emphasizes the point that there is no pattern that can be said to identify temple architecture in the Iron Age Southern Levant. More to the point, with or without unambiguous temples, there are a wide range of contexts from the Iron Age Southern Levant that suggest ritual activity and yet resist even the generic label of temple (see below).

While some scholars have argued that the heterogeneity of cultic practice of this region precludes any formal categorization (Mazar 2015, p. 30) or at the very least demands a "multi-vocalic" approach (Koch 2020), most scholars have sought to domesticate this diversity via systems of classification. We would argue that this takes us in the wrong direction, seeking to eliminate what is at the heart of religious practice in the Iron Age Southern Levant. In what follows, we will review and critique taxonomic approaches to Iron Age religion, highlighting through brief case studies architectural contexts with ritual associations that remain ambiguous within these taxonomic approaches (see Figure 1). In doing so, we will offer an alternative approach in which religion is viewed not as a distinct domain with material correlates but as one dimension of social action (i.e., engagement with the metaphysical) made visible along a spectrum of ritualization.

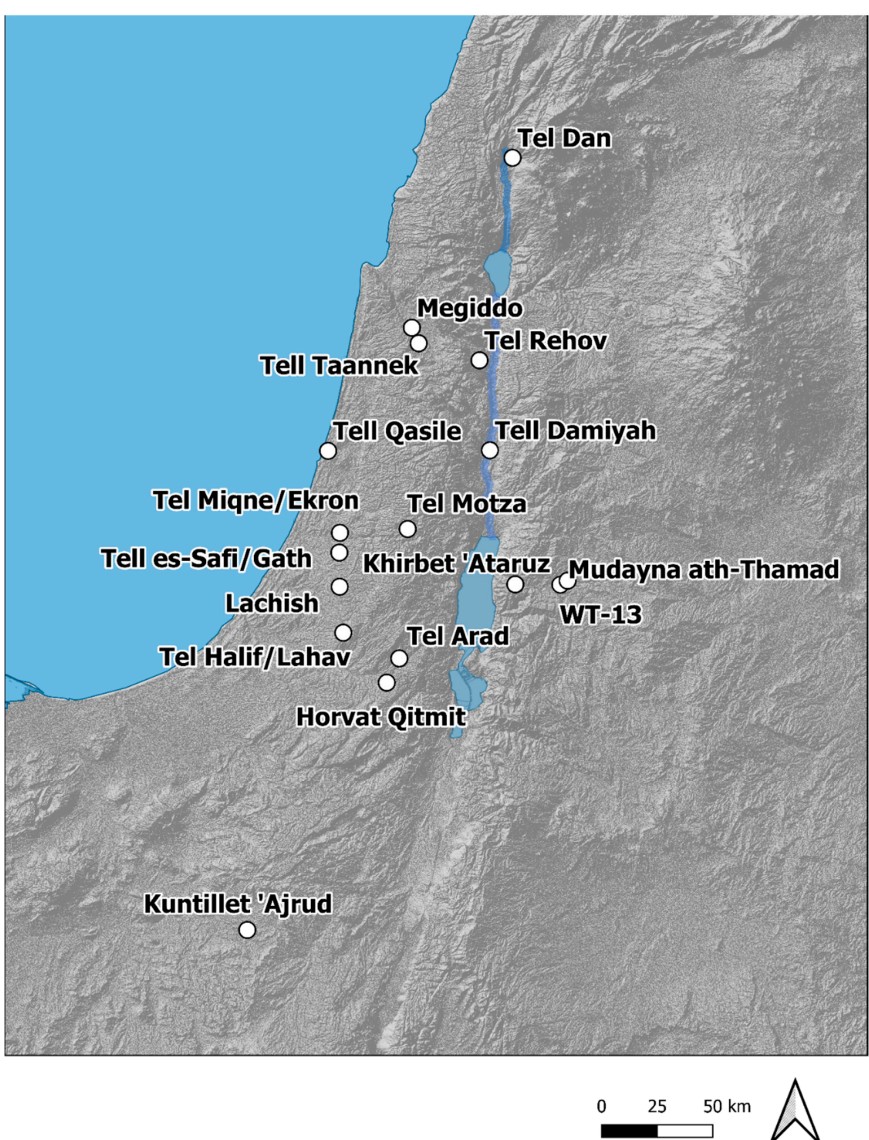

**Figure 1.** Sites mentioned in the text.

## 2. Identifying Religious Spaces

### 2.1. The Checklist Approach

From the 1980s onwards, one can find variable lists of criteria for identifying temples, cultic sites or ritual activity, in general (e.g., Lundquist 1983; Coogan 1987). Early 'check-list' approaches lacked rigorous justifications for their criteria. This changes with Colin Renfrew's seminal study of the sanctuary at Phylakopi on the island of Melos in Greece (Renfrew 1985, 1994, 2007). Renfrew (1985, p. 17) argued that " . . . structure in the belief system should engender pattern in cult practice . . . " and, hence, to list common features of sacred ritual is not a mechanical 'checklist' exercise but rather a deductive one. Renfrew supplied a list of eighteen material and behavioral correlates deduced from what he argues are the universal characteristics of sacred ritual as described in social scientific literature.

While Renfrew's work is frequently referenced in studies of Iron Age religion in the Southern Levant (e.g., Ben-Shlomo 2019, p. 3; Hess 2007, pp. 37–39; Nakhai 2001, pp. 35–36; Press 2011, pp. 369–71), it is less frequently applied in any formal sense. When it has been applied, the material/behavioral correlates have always been modified to reflect local conditions (e.g., Blomquist 1999; Daviau 2012, p. 435; Daviau 2017a, pp. 16–18; Zevit 2001, pp. 82–83). These 'necessary adjustments' highlight a number of problems both with Renfrew's approach and 'checklists' more broadly.

### 2.2. The Checklist Approach: Problems

#### 2.2.1. Affirming the Consequent

A strict application of Renfrew's system would involve the logical fallacy of affirming the consequent (Morgan 1973). Put simply, if material correlates (B) are deduced from the general characteristics of sacred ritual (A) such that the proposition "If A then B" is true, it is a fallacy to presume that one can reverse the direction of causation to say "If B then A", (i.e., to infer ritual from material remains) unless one can demonstrate that these material correlates are only and always caused by sacred ritual (see Patrick 1985).

#### 2.2.2. Equifinality and Mixed Uses

Scholars (Renfrew 1985, p. 20, pp. 362–63; Zevit 2001, p. 82) deploying these 'checklists' admit that some of their correlates may have secular as well as ritual referents. This equifinality means that an exclusive causal relationship does not exist between sacred ritual and its material correlates. Furthermore, this highlights the problematic assumption underlying 'checklists', namely that buildings, contexts and finds are generated by a single, or primary, social cause; they either are, or are not, religious in nature. This ignores the fact that apparently religious and secular activities can occur in the same spaces (Kyriakidis 2007, p. 17) and, indeed, are often ambiguously distinguished in everyday practice (Swenson 2015, p. 4).

#### 2.2.3. Inconsistent Criteria

The fact that slightly different lists of correlates have been drawn up by different scholars adapted to different regional (Zevit–Southern Levant), sub-regional (Daviau–Moab), and thematic (Blomquist–Gates) interests raises questions about how these relate to a supposedly general theory of sacred ritual. Renfrew (1985, pp. 21–22) himself sets aside domestic ritual and modifies his expected correlates in order to address it. He does this not on the basis of his general theory of sacred ritual but on the basis of his practical knowledge of domestic contexts. These correlates are inconsistent because they are not, in fact, strictly deduced from general theory but are developed contextually in light of our existing knowledge of particular sites and finds. Indeed, to be effective, 'checklists' must always be open-ended as they are in dialogue with, rather than prior to, the archaeological evidence. This suggests that even if sacred ritual were a universal phenomenon with a singular nature, its material correlates would be multiple and contextually determined.

#### 2.2.4. Post-Hoc Justification

In truth, most of points 1–3 are not really a problem because no one (including Renfrew) actually applies checklists deductively. Most checklists are not used to discover ritual sites but to justify or evaluate sites already determined to be religious on implicit grounds (Petit and Kafafi Forthcoming). Hence, checklists make the archaeologist's implicit reasoning explicit in contextual arguments of the Sherlock Holmes variety ("given this particular set of evidence B, it is reasonable to presume that A occurred"). This is known as "Inference to the Best Explanation" and is the form taken by most archaeological reasoning (Fogelin 2007). There is nothing wrong with this per se other than the misrecognition on the part of archaeologists as to what they are actually doing.

#### 2.2.5. How Many 'Ticks' Tip the Scale?

A bigger problem with the checklist approach is the goal of distinguishing religious from non-religious contexts. How many correlates, or which correlates, mark the tipping point between religious and non-religious space (cf. Steiner 2019)? Garth Gilmour (2000) attempted to address this problem in the case of the Iron Age I Southern Levant by replacing a checklist with a probability score generated subjectively by the analyst. However, what exactly does it mean to be 60% confident that a context is religious? As noted above, underlying all of these approaches is the problematic, either-or assumption that contexts are, or are not, religious in nature.

*2.3. Taxonomic Approaches*

Taxonomies are more widely deployed than checklists as a means of imposing order on the diversity of evidence from the Iron Age of the Southern Levant. Perhaps the most common typology is to divide sites according to their assumed ethno-political affiliation such that temples in Philistia, or Ammon or Israel/Judah become examples of Philistine, Ammonite or Israelite Temples (e.g., Ben-Shlomo 2019; Tyson 2019; Zevit 2001, pp. 84–85, 113–21). As both Uehlinger (2015, pp. 10–17) and Ido Koch (2020, pp. 327–28) note, these national categories are laid over the archaeological record without regard for cross-cutting similarities or internal diversity. Indeed, finding differences that map to Iron Age 'nations' usually requires a taxonomic slight-of-hand. For example, Avraham Faust (2010, 2019) is certainly correct to note marked differences between the Bronze and Iron Ages of the Southern Levant in terms of the prevalence of large-scale religious architecture. However, this difference can only be made specifically Israelite by designating structures outside of Israel and Judah, such as Building 149 at Kh. al-Mudayna or Temple 300 at Tell Qasile, as a "cultic building" or "temple" (Faust 2019, p. 6), while very similar structures in Judah, such as Room 49 from Lachish, are downplayed as " ... just cultic rooms or corners ... " (Faust 2019, p. 7) and are thus removed from consideration (cf. Zukerman 2012).

Whether or not they begin from a 'national' sub-division of the Southern Levant, many scholars go further in classifying what they see as evidence for religious activities in the built environment. However, their *taxa* are seldom generated directly from the physical attributes of this built environment. Instead, the archaeological evidence is fitted to idealized categories of religious practice that are derived implicitly, or from biblical texts, or (less frequently) from sociological models of social groupings at different scales (e.g., family, community, tribe, nation). Depending upon the scholar, these broad *taxa* are then further sub-divided in order to find places for sites already implicitly understood to be religious in nature.

The most common division is a binary, one between "state" and "popular" religion (see Ben-Shlomo 2019; Burke 2011; Dever 2005, p. 5; Holladay 1987). Temple or 'state' cult is associated with the larger, more elaborate religious architecture and presumed to have been presided over by 'official' or 'royal' authorities. Household or 'popular' cult is associated with evidence for ritual activity within domestic settings, performed by 'regular people'. This binary division creates an ambiguous space in the middle. For example, ritual activity in domestic settings is recognized by the presence of objects such as figurines (Schmitt 2014, p. 268) and miniature (model) furniture and vessels (Schmitt 2014, p. 269), or domestic shrines such as at Lahav/Tel Halif (Field IV, House 1, Str. VIB, Hardin 2010, pp. 124–60). However, it can also include 'neighborhood shrines' or 'village shrines' such as the Area E open-air sanctuary from Tel Reḥov Stratum IV (Mazar 2015, pp. 27–30). The residual body of evidence created by this binary division of religious practice has led to its critique on conceptual grounds (Stavrakopoulou 2010; Zevit 2003) and to increasingly complicated taxonomic systems to better account for the diversity of the archaeological evidence. For example, Beth Alpert Nakhai (2015) offers three domains of religious practice (individual, family, nation) each with corresponding contexts or artifacts; so too does William Dever (2005, pp. 110–75), albeit with different referents (local shrines, public open-air sanctuaries, monumental temples). Jens Kamlah (2012) also offers three categories, with eight sub-divisions, covering only temples but applied to the entire Levant from the Middle Bronze Age through the Iron Age. Zwickel's (1994) typology of cult buildings also covers the Middle Bronze Age through Iron Ages but offers five categories. Garth Gilmour (1997a) looks only at Early Iron Age evidence (Iron I-IIA) and also proposes five main types of cult sites, with three further subdivisions. Like Zwickel, Gilmour's taxonomy is hierarchical but differs in its sub-divisions. Zevit (2003) recognizes five levels of relevant social groupings in the Hebrew Bible (individual, *bêt ʾāb* ["father's house"], clan, tribe, people) each with corresponding sites of religious activity. Elsewhere, Zevit (2001, pp. 123–24) defines eight types of architectural contexts for religious activities in the Iron Age. Rüdiger Schmitt (2014) offers eight categories of Iron Age cult places with thirteen sub-categories, each linked

to distinct social groupings, which differ again from those of Nakhai and Zevit. Taken collectively, there are numerous problems in applying and interpreting these taxonomies.

*2.4. The Taxonomic Approaches: Problems*

2.4.1. Inconsistent Terminology

Not surprisingly, given the wide range of taxonomic systems, one can find many examples of the same building or complex classified in different ways.

For example, the cultic building at Tel Arad has been classified as a monumental temple, a local temple (both in Dever 2005, pp. 167, 175), a regional sanctuary (Schmitt 2014, p. 275), a fortress sanctuary (Kamlah 2012, pp. 508–9), an intimate temple for popular worship (Herzog 2002, p. 68), and most recently as a shrine, rather than a temple, "due to its modest dimensions" (Arie et al. 2020, p. 5n.1). Locus 2081 at Megiddo has been designated as an establishment shrine (Holladay 1987, p. 271) or a cult room in an administrative building (Kleinman et al. 2017, p. 41). It has also been designated as a domestic shrine (Schmitt 2014, p. 269) providing " … evidence of the religion of a *bêt ʾāb*" (Zevit 2003, p. 233). These terminological inconsistencies are important because they potentially change the social collective or categories of religious practice to which the building is being associated. Since these categories constitute the explanation for the existence of the building in the taxonomic approach, changing categories changes meanings.

2.4.2. Variability within Taxa

The taxonomic approach operates on the assumption that its categories are comprehensive and mutually exclusive. Everything needs to be placed in a category whose members are presumed to share essential attributes. This downplays diversity within categories, something that is exceptionally common in the case of the Iron Age Southern Levant. For example, in summing up the buildings he has classified as temples, Ziony Zevit (2001, p. 254) states: "What is remarkable about those whose general layout is clear … is how much they do not resemble each other … ." In this sense, adhering to strict and static taxonomic categories can oversimplify past human activity, and, by extension, the societal mechanisms we aim to investigate.

2.4.3. Similarity between Taxa

Another by-product of being comprehensive and mutually exclusive, is that taxonomies downplay similarities that cut across categories. John S. Holladay's (1987, pp. 280–82) assertion that one could identify a clear division between aniconic "Establishment" practices and iconographic "Non-conformist" practices has not stood up to subsequent evidence. Instead, we see many cross-cutting features in identified Iron Age religious contexts. For example, looking at Schmitt's (2014, pp. 279–81) elaborate classification chart, we see that eleven out of his thirteen sub-types of cult-places include contexts with offering stands, ten include contexts with votive figurines, ten include contexts with chalices, eight include contexts with benches and all thirteen include contexts with utilitarian vessels used for food and drink consumption. The variable ways in which these common attributes are rearranged and realized across taxonomic categories cannot be captured within these taxonomies.

2.4.4. Taxa as Non-Explanations

Ritual activity is characterized by its fluid and dynamic behaviors (Insoll 2011, p. 3). In the taxonomic approach fluidity and variability can only be captured by adding more *taxa* and/or restructuring the taxonomy at regular intervals. One can see this in the proliferation of classification systems that we reviewed above. One can also see this in the uncertainty attached to the assignment of buildings or sites to specific *taxa*. As one scholar notes of his own taxonomy " … it is often not easy or even possible to determine to what sub-category each shrine belongs" (Gilmour 1997a, p. 6). These taxonomies purport to both organize and explain the archaeological contexts they contain (e.g., "this building is a neighborhood shrine *because* it served the religious needs of a *bêt ʾāb*"). However, if

individual archaeological contexts are ambiguously classified or regularly reclassified, or if the *taxa* themselves change between classification systems, one might question how much these taxonomies actually explain.

### 2.4.5. Taxonomic Tunnel Vision

As in the case of checklists, the purpose of taxonomies is to identify religious spaces. Even when quotidian activities are central to the identification of specific categories of religious space, such as "Domestic" or "Industrial" cults, those activities are withdrawn from consideration except insofar as they can be viewed as engaged with, or expressive of, religious activities. This segments spaces conceptually that are unified in reality. An interesting exception to this trend is the attempt to make Iron Age women visible in the archaeological record using feminist frames of reference (see Nakhai 2019). This work has focused on domestic labor as gendered and inclusive of religious activities, a holistic approach which informs the final section of this paper.

### 2.5. The Spectrum Approach

Checklists and taxonomies address a certain anxiety over the ascription of religious meanings or functions to buildings and artifacts. This anxiety can be found early in the history of biblical archaeology (May 1935, p. 1), as can attempts to use critical and explicit methods as a solution (McCown 1950). This anxiety, and the stated need for formal classification, has often been expressed in terms of a contrast between the metaphysical referent of religion, manifest in beliefs, values, and worldviews, and the material referent of archaeology, manifest in artifacts, features and buildings (Insoll 2004, p. 154). However, as the so-called "material turn" in the study of religion has emphasized (Bräunlein 2016), religion itself is a material domain that involves immanence as well as transcendence (Reinhardt 2016). The common practice in the archaeology of religion of mapping religion to immaterial belief and ritual to concrete action perpetuates a mind–body dualism that misrepresents both thought and action. The affordances, agencies and semiotics of places and objects, the effects of bodily dispositions and sensorial experiences, do not merely supplement disembodied belief. From a Material Religion perspective, engagement with the metaphysical and the physical are co-constitutive; hence, religion is realized in, and not abstracted from, a lived material setting (Keane 2008). The problem, therefore, is not the existential question of whether an immaterial religion leaves material traces behind, but rather the archaeological question of following " ... the same trail but walking it backwards" from material traces to religious activities (Steiner 2019, p. 1). As Steiner notes, most work in the archaeology of religion has walked this trail forward from presumed religious activities to material correlates as if the path were singular and led in only one direction. We would add that walking this trail backwards also raises the question of whether material remains always lead to a singular destination, one that either is, or is not, religious in nature.

We would propose an alternative path that approaches religion contextually in terms of recognizable activities understood to be fluid and dynamic in their meanings and referents. Such a perspective enables us to discuss evidence for ritual interaction with other forms of human activity, facilitating a more holistic discussion of evidence without forcing the archaeological realities into inflexible molds. The past significance of a given context can be seen to fall on a multidimensional spectrum of human activity. Religion, as engagement with the metaphysical, can form one dimension of said spectrum, intersecting with a multitude of other human activities. A spectrum is, by definition, a range of different positions between two extreme points. Between these points fall all possible variations.

Rather than fret over defining spaces as religious or secular, we take engagement with the metaphysical to be a constant (i.e., religion is always there). This is not to revive some version of mythopoeic thought or to deny a premodern sense of practical causation. What we propose is closer to the role of the microscopic as a constant in our current world. We accept microscopic phenomena as a constant presence, but they are only brought

to our attention irregularly and by a variety of informal and formal means, mediated by different contexts, levels of specialist knowledge, social roles and sets of priorities. Saying "I think I have caught the flu bug" is different from working as a microbiologist in a laboratory or receiving a COVID-19 vaccination, but all three are moments in which the microscopic is made visible. In the case of religion, we would follow Catherine Bell (1992) and describe the different ways in which engagement with the metaphysical is made visible as 'ritualization'. Rather than define the constituents of 'ritual' in universal terms, Bell focused on ritualization as the process by which some activities are marked off and privileged in relation to the everyday.

> "As such, ritualization is a matter of various culturally specific strategies for setting some activities off from others, for creating and privileging a qualitative distinction between the 'sacred' and the 'profane', and for ascribing such distinctions to realities thought to transcend the powers of human actors" (Bell 1992, p. 74).

We would add that ritualization occurs along a spectrum with greater or lesser degrees of distinction, separation and privilege in relation to everyday life. Hence, ritualization makes engagement with the metaphysical more or less visible, especially from an archaeological perspective. This spectrum of ritualization is significant because it strategically creates what Bell calls "ritualized bodies" who occupy different social positions with different potentialities and powers within a larger set of recognised ritual practices.

From an archaeological perspective, this allows us to reorient our questions from "is this ritual?" to "what is being ritualized, to what degree, by what means and with what effect?" These are inherently contextual questions and Bell is adamant that " . . . ritualization can be characterized in general only to a rather limited extent since the idiom of its differentiation of acting will be, for the most part, culturally specific" (Bell 1992, p. 93). However, in the case of the Iron Age Southern Levant, the three common qualities of ritualization that Bell cites, " . . . formality, fixity and repetition" (Bell 1992, pp. 90–91), work very well. Other possibilities include elaboration, iconicity (or aniconicity), sensorial enhancement, isolation/singularity, cost, scale (e.g., monumentality or miniaturization) and quantity.

### 3. Between Rocks and 'High Places': Applying the Spectrum Approach

Moving beyond checklists and taxonomies, we want to consider the outliers that these approaches create. These outliers allow us to illustrate more concretely the spectrum along which ritualization makes engagement with the metaphysical visible.

*3.1. Wayside Shrines?*

3.1.1. Tell Damiyah

Tell Damiyah, located in the central Jordan Valley, is a small tell-site (2.9 ha, <1 ha summit) which overlooks the confluence of the Jordan and Zerqa Rivers, as well as one of the few fords over the Jordan River (Petit and Kafafi 2016, p. 19). Excavations in Stratum VII on the summit of the site revealed the remains of several buildings that were destroyed by a conflagration around 700 BCE. This conflagration resulted in several sealed contexts, providing valuable evidence of how the buildings were used prior to their destruction.

The remains of two clusters of buildings were encountered (see Figure 2a). The southern cluster, consisting of two separate rooms (ca. 3 × 2.5 m), yielded evidence for domestic activities: cooking pots and mortars and pestles pointed to food preparation, whereas loom weights indicated textile production. The northern cluster consisted of a single building containing two rooms, spanning an area of ca. 10.6 × 4.2 m. Little in its architectural layout, except perhaps its size (compared to mudbrick buildings in this region) and the presence of a mudbrick platform in each of the rooms, would signal a strictly 'religious' purpose (Halbertsma Forthcoming). It is only when looking at the finds within and adjacent to the building that the evidence becomes convincing. These finds include: two ceramic anthropomorphic statues found in the passageway in front

of entrance to the structure, two bovine skulls placed on the floor opposite the entrance, numerous zoomorphic and horse-and-rider figurines, a ceramic offering stand and the head of a third ceramic statue, all within the structure. It is important to note that all of the female figurines from the site, including those ascribed to Stratum VII in initial publications (Petit and Kafafi 2016, Figs. 16–17) have now been reassigned to the earlier and only partially exposed Stratum VIII. These finds and the small size of the site led the excavators to interpret the building as an isolated sanctuary (Petit and Kafafi 2016). However, recent excavations directly north of the 'sanctuary' have revealed that this building was, in fact, much larger than previously thought. Several adjoining rooms to its north yielded evidence for large-scale storage of foodstuffs and possibly redistribution (Petit and Kafafi Forthcoming). Hence, it is now clear that religious activity was one of several recognisable activities carried out in this building, possibly linked by a common institutional setting (Halbertsma Forthcoming).

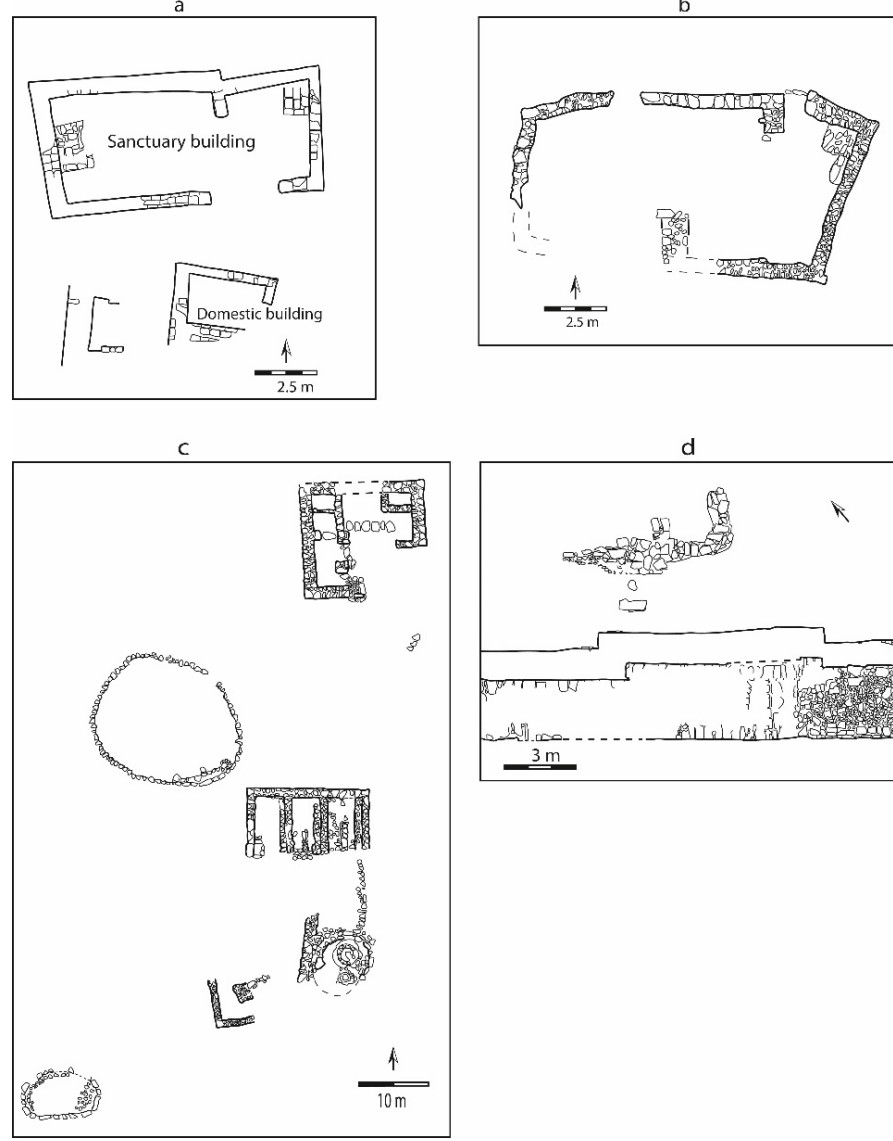

**Figure 2.** (**a**): Tell Damiyah Phase VII (after Petit and Kafafi 2016, Fig. 5); (**b**): WT-13 Stratum IIB (after Daviau 2017, Fig. 3.26); (**c**): Horvat Qitmit (after Beit-Arieh 1995, Fig. 1.6); (**d**): 'Ein Haseva favissa (after Ben-Arieh 2011, Plan 1).

Ritualization in the case of Tell Damiyah is not invested in to any significant degree in architecture. The size of the 'sanctuary' and perhaps also the platforms are in keeping with the apparently non-domestic nature of the building but only become correlates of ritual in light of the finds. In contrast, energy was invested in setting votives and offerings apart from everyday objects, making it clear that this is where ritualization is concentrated. Cost and singularity mark the deposit of bucrania, which is unusual for the Iron Age Southern Levant. Iconicity is the most obvious quality of the ceramic statues and figurines. However, the recognizable types represented by the 'horse and rider' figurines constitutes a kind of formality that references standardized and widely circulating forms. The ubiquity and continuity of simple fenestrated offering stands such as the one from Tell Damiyah also mark a notable ritualization through formality, both in the conservation of a distinct form over millennial and in the structured actions of offering that the stands suggest.

### 3.1.2. Other Sites with Statues

In this same way, the ceramic statues (essentially pot-like cylinders with attached anthropomorphic features), connect Tell Damiyah VII to a smaller number of sites where these relatively rare objects are found; namely, WT-13 Stratum IIB-A (Daviau 2017a) in central Moab, and Ḥorvat Qitmit (Beit-Arieh 1995) and 'Ein Ḥaṣeva (Cohen and Yisrael 1995; Ben-Arieh 2011) in the Negev. All of these sites are situated along significant travel routes in transitional zones between temperate and steppic environments or between valleys and plateaus. At the same time, these sites are also all distinct in their architecture and assemblages (see Daviau 2012, 2017a, pp. 275–77). WT-13 is a lone rectangular structure (see Figure 2b) that was probably not fully roofed located some 250 m from the fortified settlement of ar-Rumayl in central Moab. Ḥorvat Qitmit is an isolated collection of buildings, pens and open-air features in the north-eastern Negev (see Figure 2c). 'Ein Ḥaṣeva is a fortress in the eastern Negev with a pit (*favissa*), containing the statues and related ritualized artifacts, associated with the enigmatic remains of a largely disassembled structure (see Figure 2d). The pit and enigmatic architectural remains were located outside of the fortress adjacent to the northern perimeter wall. All of the sites date to the Iron Age IIC period (7th–6th C. BCE), except for WT13-Stratum IIB-A, which dates to Iron IIB (c. 830–700 BCE).

WT-13 yielded the largest number of whole or partial statutes and shows the widest range of forms (Daviau 2017a, pp. 108–28). Some of the WT-13 statues are open on both ends and could function as offering stands (i.e., they could hold bowls on their heads), at least one had a lamp attached to its head, others have fully closed and rounded heads but carry model offerings in their hands and some are empty handed. All of the Negev statues appear to be offering stands, while the one head from Tell Damiyah is closed and rounded and hence could not have held a bowl. The other two statues from Tell Damiyah are only preserved from the neck down and are not carrying offerings. These statues represent an elaboration of the standard offering stand through the addition of iconicity. At the same time, the Transjordanian examples represent a further innovation if the statues with closed heads are understood as representations of worshippers making offerings.

Michèle Daviau's (2017a, p. 127) thorough review of parallels for the statues noted that further whole or fragmentary examples have been published in small numbers from sites in Moab, Ammon, Northern Jordan and Syria, raising questions about the standard 'ethnic' explanation for these objects. After all, in what sense is 'Transjordanian' an ethnic identity? The different sample sizes make it difficult to know if all sites had the same range of statue forms as at WT-13, or whether this particular ritual practice was realized differently in different contexts. The same could be said about these sites overall. All could be covered by a category such as "wayside shrine", but this masks significant differences. The lack of contextual information for the 'Ein Ḥaṣeva finds makes it difficult to interpret. WT-13 seems to be almost wholly dedicated to religious activities, whereas Tell Damiyah has considerable evidence for everyday production, even if much of this turns out to have been organized within a 'religious' institution. Ḥorvat Qitmit is more difficult, but we

favour Zevit's (2001, pp. 143–48) observation (against those of Beit-Arieh 1995, pp. 21–26) that Complex B and many of the circular pens do not differ from contemporary sites from across the Negev and hence should be interpreted in terms of animal husbandry and domestic occupation. Certainly, specific ritual practices differed to some degree—both 'Ein Ḥaṣeva and Ḥorvat Qitmit contain offering stands and chalices with a range of zoomorphic, anthropomorphic and botanical attachments, whereas these are missing from WT-13 and Tell Damiyah. WT-13 and Ḥorvat Qitmit contain architectural models which are missing from the other two sites. Tell Damiyah and Ḥorvat Qitmit contain zoomorphic figurines, whereas WT-13 contains a large number of female figurines and very few zoomorphic figurines, while the 'Ein Ḥaṣeva favissa did not contain any figurines. Small stone altars are abundant at 'Ein Ḥaṣeva, represented by one example at WT-13 and unattested at Ḥorvat Qitmit and Tell Damiyah. Overall, what we can say is that these sites shared liminal positions in terms of travel routes, and also shared a partially overlapping offering cult, but were set in distinct architectural and institutional settings.

### 3.1.3. Kuntillet 'Ajrud

Another site that has been classified as a "wayside shrine" is Kuntillet 'Ajrud. Located on a hilltop overlooking the Wadi Quraiya in the north-eastern Sinai, Kuntillet 'Ajrud is a short-lived site dating to the early Iron Age IIB (ca. 830-750 BCE, Carmi and Segal 2012, p. 61; cf. Finkelstein and Piasetzky 2008, p. 184). The site comprises two buildings (A and B), both built of stone and partially preserved on the surface (Meshel 2012, p. 13). While Building B had undergone substantial weathering and was more difficult to reconstruct, Building A clearly showed the outline of a rectilinear building, measuring ca. 29 × 15 m, with possible towers on each of its corners (see Figure 3). The plan of the building most closely resembles that of Iron Age fortresses such as Tell el-Qudeirat/Kadesh Barnea (Cohen and Bernick-Greenberg 2007) and several Negebite fortresses (e.g., Cohen 1979, Fig. 5.3). There were two benched rooms near the entrance of Building A, indicative of places for public gathering.

Kuntillet 'Ajrud has been variously interpreted as a religious school (Lemaire 1981, pp. 25–30), a religious site near a 'sacred grove' (Na'aman and Lissovsky 2008), a cult complex (Zevit 2001, pp. 374–81), a wayside shrine on an Arabian trade route (e.g., Finkelstein 1992, p. 163; Meshel 2012, p. 54), a fort with a gate shrine (Dever 2005, p. 160) and a caravanserai (Hadley 1993). The association of the site with religious practice stems mainly from a collection of inscriptions and images painted on two pithoi and fragments of wall plaster. Most famous are its two inscriptions mentioning "*Yahweh of Samaria/Teman and his 'asherah*" (Aḥituv et al. 2012). The furor over the association of Yahweh and Asherah in these inscriptions has distracted attention from the actual material remains of the site. The entryway and adjacent benched rooms of Building A were both plastered with murals and contained the two pithoi with painted imagery and inscriptions. This has led some scholars to describe the benched rooms as spaces dedicated to religious activities. At the same time, the site lacks the typical votives and accoutrements associated with offering cults.

Tallay Ornan (2016) has argued convincingly that the imagery of both the murals and the pithos replicate themes associated with kingship and apotropaic protection. She sees the site as a royal military establishment adjacent to the *Darb al-Gazza*, the route from Gaza to the Red Sea. Smoak and Schniedewind (2019) agree and argue that the skilled execution and formulaic nature (e.g., blessings as an epistolary greeting formula) of these inscriptions show them to be scribal exercises, whose religious content simply illustrates the embeddedness of metaphysical language in Iron Age society. Whether exercises or primary texts, both the imagery and the texts do engage with the metaphysical visible and, hence, constitute some level of ritualization. However, rather than marking metaphysical engagement off from other aspects of life at Kuntillet 'Ajrud, these literary formulae and iconographic tropes simply highlight the embeddedness of religion in royal ideology and scribal culture. Overall, we can say that although the architecture of the site shows only

limited evidence for ritualization, Kuntillet 'Ajrud does show that religion was entangled with many aspects of the site. However, this does not take the form of a clear offering cult, nor can the site be considered primarily religious in function. As we noted, engagement with the metaphysical is a constant but one that is made visible to different degrees in everyday life. Asking "what is being ritualized, to what degree, by what means and with what effect?" in the case of Kuntillet 'Ajrud leads us not to a highly ritualized offering cult, but rather to the entanglement of religion with royal ideology and scribal practice.

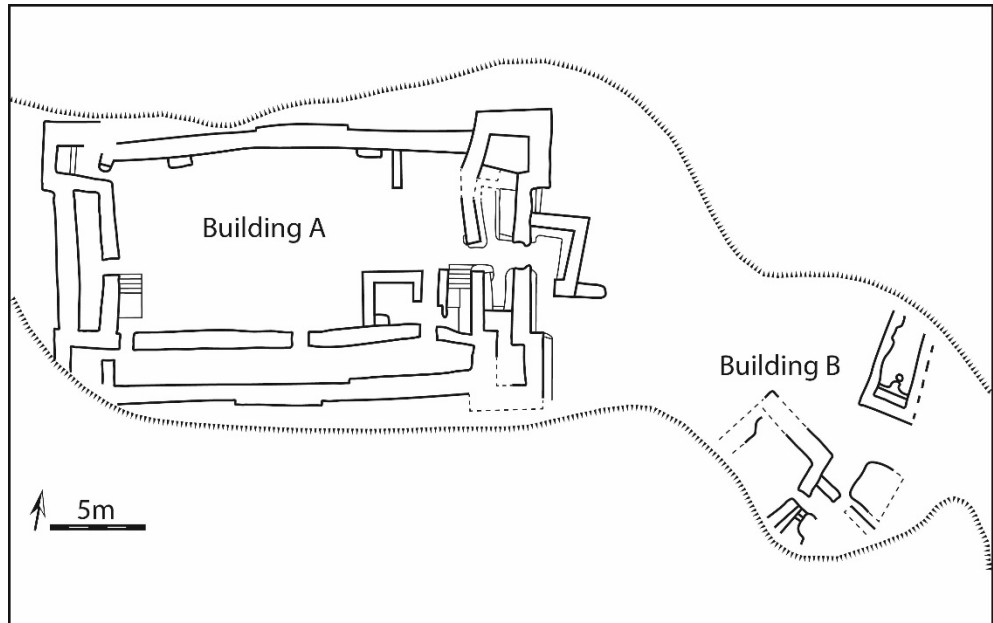

**Figure 3.** General plan of Kuntillet 'Ajrud (after Meshel 2012, Fig. 2.1).

### 3.1.4. Diversity within Categories

Treating "wayside shrines" as a single phenomenon does not help us understand what was going on at each of these sites. Instead, paying careful attention to how Tell Damiyah, WT-13, 'Ein Ḥaṣeva and Horvat Qitmit differ in terms of their activities, ritual practices and likely institutional setting is key to highlighting more precisely where they are similar. In the case of Kuntillet 'Ajrud, these differences highlighted what was missing, namely an offering cult.

### 3.2. *Industrial Cults?*

More ambiguous than the statues and elaborate stands of 'Wayside Shrines' is the mix of ritualized and quotidian finds that have led some scholars (Daviau 2014; Schmitt 2014) to define 'Industrial Cults' as a type of religious context in the Iron Age of the Southern Levant. We will consider several of these in order to explore the ambiguous border between the 'sacred' and 'profane' and role of ritualization in marking or eliding this border.

### 3.2.1. Tel Reḥov Apiary

Excavations at Tel Reḥov, located in the Beth-Shean Valley, uncovered the remains of a unique apiary in Area C dating to the Iron IIA Stratum V (late 10th/early 9th century BCE). The apiary consisted of an unroofed space enclosed on three sides and three rows of unbaked clay cylinders stacked in three tiers, each housing a hive. The excavators uncovered 30 cylinders and estimate an original total of c. 180 hives, indicative of a well-organized and industrial scale undertaking (Mazar 2018, pp. 41–42; Mazar and Panitz-Cohen 2007, p. 207). Just over six meters south of the rows of hives, excavators found a four-horned fenestrated clay altar decorated with two applied female figures separated by an incised tree (Mazar 2015, p. 31; Mazar and Panitz-Cohen 2007, pp. 209, 212). Adjacent to

the altar, excavators found a large and elaborately decorated chalice with applied petals (Mazar and Panitz-Cohen 2007, p. 212). At least three, and perhaps four, other ceramic four-horned altars (Mazar 2015, Figs. 3–4) have been found in unusual Iron IIA contexts at Tel Reḥov (Area E "Open Air Shrine" and buildings CF and CP). As Amihai Mazar has shown (Mazar 2015, pp. 32–36), these ceramic altars should be considered in conjunction with other rectilinear, fenestrated and decorated artifacts that have at times been classed as 'cult stands' and sometimes as 'architectural models' (see Katz 2016). It seems clear in the Reḥov examples that both the altar and the chalice are related to the presentation of offerings, perhaps not always burnt as only the Area E altar contains soot marks (Mazar 2015, n. 18). This small-scale offering cult has been made visible (ritualized) by the elaboration and formal qualities of its accoutrements rather than its isolation or architectural setting. Indeed, the intent seems to be to integrate the offerings and the honey/beeswax production in a relatively seamless way.

### 3.2.2. Khirbat al-Mudayna, Building 200

The integration of ritual activity and economic production appears to have been widespread in the Iron Age of the Southern Levant. The site of Khirbat al-Mudayna on the Wadi ath-Thamad in east-central Moab (Jordan) has provided a range of evidence for ritualized activity. This is most obvious in the case of Building 149, with its restricted space, benches and three limestone altars (Daviau and Steiner 2000). A less obvious case is the Building 200-205-210 complex located south of Building 149 (Daviau et al. 2012, Fig. 20). These three buildings share partition walls, pillared interior walls and each measures c. 12 × 6–7 m (see Figure 4). In plan these buildings share the layout of either a four-room house (Building 200 and 205) or a tripartite pillared building (Building 210). However, they are understood by the excavators to be a specialized industrial complex dedicated to weaving (Daviau and Chadwick 2007). Based on the distribution and morphology of loom weights, Jeanette Boertien (2013, pp. 230–31) reconstructs four looms in this complex plus one more in the adjacent Building 145. Other weaving-related finds include spindle whorls, bone spatulas, linen and wool textile fragments and metallic ores that may have served as dye pigments. Limestone basins were found both plastered in between pillars and displaced elsewhere across the complex. Although bins formed by low walls in between pillars are a common feature in four-room houses, the substantial limestone basins in Buildings 200 and 210 are more reminiscent of the limestone basins found in between pillars in the Stratum IVA Tripartite Pillared Buildings at Megiddo. These structures have been interpreted in a variety of ways, but in the case of Megiddo are most commonly interpreted as royal stables with feeding troughs (Cantrell 2011). This may support the public rather than domestic interpretation of the Khirbat al-Mudayna buildings. Boertien (2013, p. 293) suggests that these basins may have played a role in dying textiles or fulling wool. Other unusual finds include several rectangular limestone tables or stools that Daviau (2017b) suggest may have served as seats for textile workers such as those depicted in Egyptian tomb paintings.

While accepting the non-domestic nature of this complex, Boertien (2013, pp. 232–33) argues that the scale of production indicated by the looms is insufficient to support a specialized export industry. Peter Popkin (2009, pp. 239–41), in a study of faunal remains, concluded that herd management strategies were oriented towards the production of wool but within a context where herd security was prioritized and animal products were distributed locally. Hence, the issue seems to be a question of the scale of production rather than the existence of specialized production.

In addition to weaving related finds, grinding implements and a large pottery assemblage described as "utilitarian" (Daviau 2014, p. 122), c. 17 medium and miniature stone altars were discovered in this complex and the adjacent street (Daviau 2007, 2014, pp. 119–23). These altars appear to have been used for burning aromatic plant material (rather than resins) (Daviau 2014, p. 122). A large cache of "hundreds" (Wadi ath-Thamad Project 2020) of astragali were also found here in Room 201 of Building 200 (Daviau et al.

2006, p. 258). Boertien (2013, pp. 285–93) suggests that, while not producing textiles for export, Building 145 and the Building 200-205-210 complex may have been engaged in the production of textiles for the Building 149 'temple'. Boertien notes a number of contexts in the Southern Levant where loom weights and cultic activities seem to be related. She also notes that a number of these contexts also contain caches of astragali, leading her to wonder if astragali might have been used as a tool in weaving such as a shuttle (Boertien 2013, pp. 292–93). Experimental work has shown that astragali from large ruminants (cattle and deer) could have functioned as loom weights or as spools (Grabundžija et al. 2016) as the shape allows the easy and secure attachment of thread. However, the caching of astragali is a well-known phenomenon in the Levant and Eastern Mediterranean. While by no means exclusively found in ritualized contexts, this is one of the more common places to find large caches (Gilmour 1997b; Susnow et al. 2021). When studied carefully, these caches often show side preferences (rather than left/right pairing) and intentional modification (see below) that might actually inhibit the attachment of thread. A variety of textual and iconographic sources suggest that, while astragali could have had a variety of uses, they were often used for casting lots in both games of chance and divination (Gilmour 1997b; Susnow et al. 2021).

Quite rightly, Daviau has connected the industrial contexts of the altars to the similar stone altars found in the industrial quarter at Tel Miqne-Ekron, amongst large numbers of both olive presses and loom weights (Gitin 1989, 1992). However, the astragali cache also links the Building 200-205-2010 complex to a number of other interesting contexts.

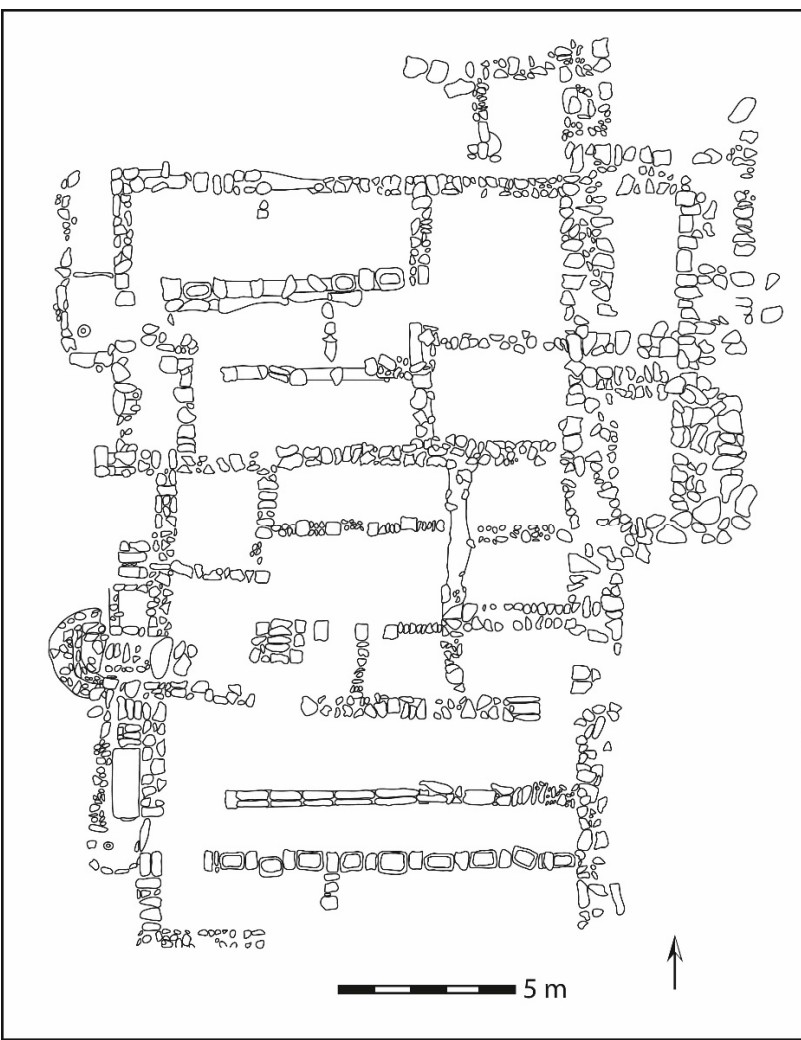

**Figure 4.** Khirbet al-Mudayna Building 200, B205 and B210 (after Daviau et al. 2012, Fig. 20).

### 3.2.3. Taanach "Cultic Structure"

The so-called "Cultic Structure" from Period IIB (late Iron IIA) at Taanach contained loom weights, three caches of astragali, as well as evidence for metal production. Three offering stands were found in the immediate vicinity of these structures, two by Paul Lapp in Cistern 69 (Lapp 1969, pp. 42–44; Rast 1978, Fig. 54) and one by Ernst Sellin (1904, pp. 75–79) with an imprecise findspot. Two of these stands are quite famous, elaborate rectangular stands, sometimes labelled as architectural models (Katz 2016), and decorated with three-dimensional anthropomorphic and zoomorphic figures. Two chalices were also found in Cistern 69 (Rast 1978, Fig. 53).

The interpretation of the "Cultic Structure" has long been controversial (Yeivin 1973, pp. 172–73; Fowler 1984; Rast 1994; Zevit 2001, pp. 235–37). In part, this is because of the disturbed nature of the area and the difficulties in coordinating the results of Sellin's excavations in 1902–1904 and those of Lapp between 1963 and 1968. However, it is also a by-product of the assumption that the "Cultic Structure" must be one or the other, religious or secular. What survives of the architecture seems domestic or industrial in nature. Indeed, Frank Frick (2000, pp. 52–53) restores the partial plan of the structure (see Figure 5) in the form of a four-room house (albeit problematically due to the position of Basin 75). In the two surviving rooms, Lapp found large quantities of pottery (Rast 1978, Figs. 30–50), as well as grinding implements, a large krater with 62 whole or fragmentary clay loom weights inside (Friend 1998, pp. 43–58; Frick 2000, pp. 129–32), and two bone spatulas that likely functioned as weaving tools (Friend 1998, pp. 63–65; Frick 2000, p. 143). Glenda Friend (1998, p. 10) suggests that these weights could have supplied three or more looms. The density of finds in Room 1 has led most scholars to view it as storage space. Other 'industrial' finds associated with the "Cultic Structure" included a large number of iron tools and weapons (Frick 2000, pp. 148–59) as well as tuyère fragments, copper spillage and copper ore, suggesting casting and perhaps smelting in the vicinity (Stech-Wheeler et al. 1981, p. 249).

Scholars (Yeivin 1973, pp. 172–73; Fowler 1984; Zevit 2001, pp. 235–37) have questioned the cultic interpretation of this structure because (1) there is a stratigraphic disconnection between the surviving architecture and the findspots of the offering stands; (2) the Room 1 assemblage and the architecture of the structure seem to be domestic in nature; and (3) Lapp (1964, p. 6) set out to recover cultic materials in this area because of Sellin's discovery of the offering stand. However, Room 1 is not entirely devoid of ritualized objects. These finds include a figurine mold (Frick 2000, pp. 108–14), a small fenestrated stand (Rast 1978, Fig. 51: 4) and three clusters of primarily sheep and goat astragali totaling c. 125 whole and 22 fragmentary examples (Frick 2000, p. 71). At least 15 of these astragali showed signs of working, including cutting, polishing, drilling and the insertion of metal pegs (Frick 2000, pp. 74–75). Intentional modification is not uncommon in astragali caches and may relate to improving their performance as dice or their ability to signify in divination (Susnow et al. 2021). Walter Rast (1994) argues for a middle position in which the "Cultic Structure" served as storage space for priestly families. In our view, much of this debate is generated by taxonomic approaches that label built space as either religious or secular, rather than asking the questions "what is being ritualized, to what degree, by what means and with what effect?" We will consider one further example before looking at these issues more closely.

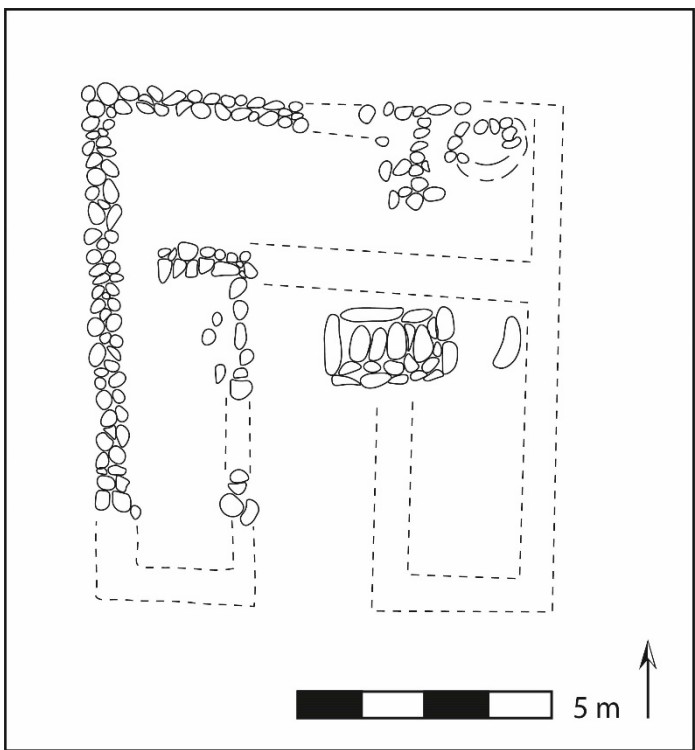

**Figure 5.** The "Cultic Structure" from Tanach (after Frick 2000, Fig. 3).

### 3.2.4. Tell eṣ-Ṣafi Area D Temple

In Area D, on the north side of the lower city, at Tell eṣ-Ṣafi a sequence of buildings was uncovered that has been interpreted by the excavators as superimposed temples (see Figure 6). The buildings of Strata D4 and D3 span the late Iron IB and Iron IIA periods (10th–9th centuries CE). Both have features associated with domestic architecture; the D4 building has an internal row of pillars like a three-room house, while the D3 building has a layout similar to a four-room house. The D3 Building (No. 149807) is the more extensively excavated, measuring c. 11 × 15 m and associated with a metal working area immediately to the east. Building 149807 contained a large and unusual two-horned stone altar set into the south-east corner of a narrow room near the entrance to the building (Dagan et al. 2018, p. 31, Fig. 4). Finds from this building included a large number of elaborate painted chalices, some with attached petals (Dagan et al. 2018, p. 31, Fig. 5), over 200 astragali (Dagan et al. 2018, Fig. 6) and ca. 250 loom weights in several clusters (Cassuto 2017, p. 193) that suggest they had been in storage at the time of the destruction of the building (Cassuto 2018, p. 57). Immediately east of Building 149807, the remains of a metal-working workshop were discovered (Workman et al. 2020). Here, some 46 kg of production waste, such as iron slag, crucibles, tuyères and vitrified ceramic debris were recovered (Workman et al. 2020, p. 213). A small room (20D06A05) accessed from the workshop contained a further 40+ loom weights, chalice fragments and a "standing stone", indicating that ritual activity and weaving were not limited to Building 149807 proper.

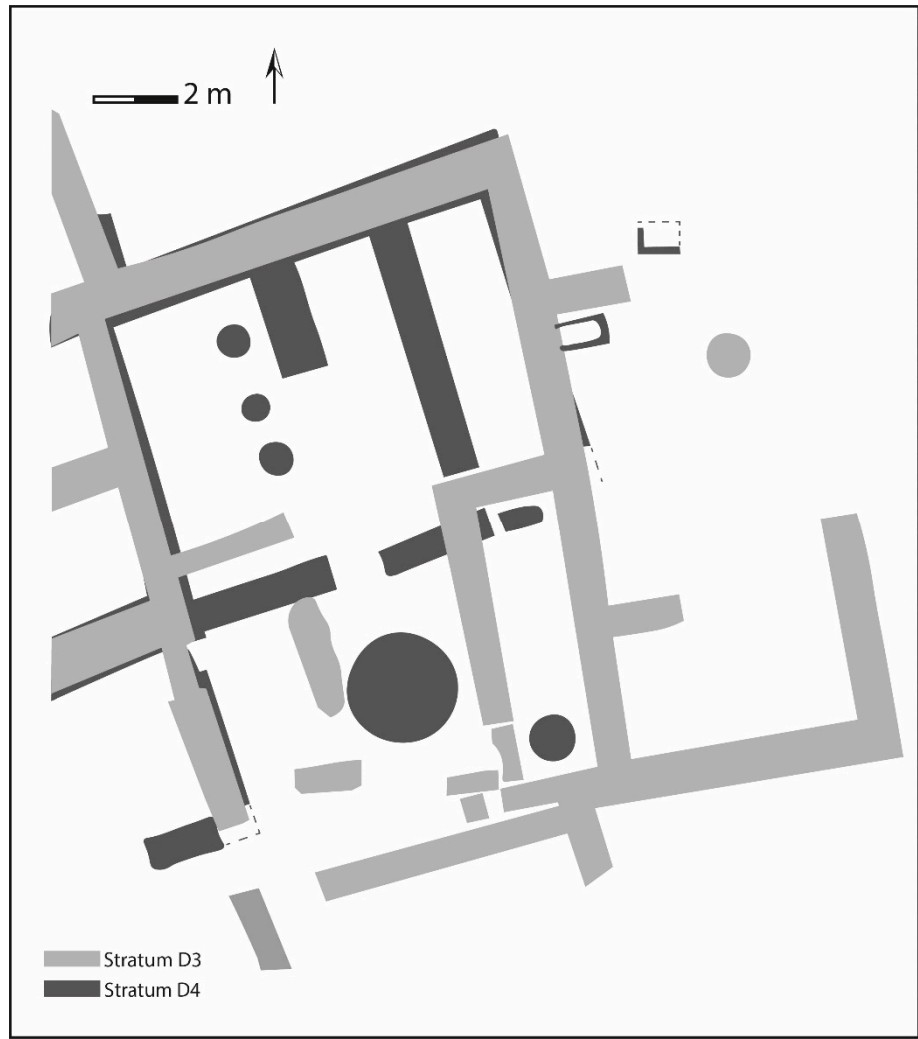

Stratum D3
Stratum D4

**Figure 6.** Tell eṣ-Ṣafi Area D Temple (after Dagan et al. 2018, Fig. 2).

### 3.2.5. Similarity across Categories

The Tel Rehov Apiary and Building 200 at Kh. al-Mudayna might be grouped together as industrial cults since offering-related artifacts are secondary to the evidence for economic production in each context. At the same time, Building 200 shares a larger complex of finds with structures that tend to be placed in distinct categories—the Taanach "Cultic Structure" is classified as priestly/cultic storage, while Building 149807 at Tell eṣ-Ṣafi has been labelled as a temple. All three share some architectural attributes, but these attributes are normally associated with domestic architecture. Hence, we cannot resolve this pattern with a cultic label such as 'Temple Production'. Rather, divination and offerings seem to have been part of the process of weaving, at least when carried out on a larger scale outside of the home. Making things and making offerings were linked activities that shared the same architectural spaces in the Iron Age of the Southern Levant. This is only problematic if we start from the position that religious and secular spaces are categorically distinct.

### 4. Conclusions

It may seem perverse to contribute to a special issue on religious architecture without offering any clear examples of religious architecture. However, the spectrum approach that we have proposed does allow us to think differently about architecture and its relation to religious practice. If we take engagement with the metaphysical as a constant and focus instead on where, how and to what degree it is made visible by ritualization (i.e., being

set apart), religious architecture becomes not a set of universal attributes but one of many strategies of ritualization.

As our case studies show, focusing on strategies of ritualization changes how we look at religious practice. Rather than debating whether contexts were religious or redefining our *taxa* of ritual contexts in the face of new evidence, we were able to highlight more precisely where, and to what degree, effort in ritualization was invested. In the case of "Wayside Shrines", we highlighted the common phenomena of ceramic statues in the offering cult while still identifying diversity between sites. Rather than accounting for the cooccurrence of 'sacred' and 'secular' activities by inventing new categories such as 'Industrial Cults', we were able to explore common forms of ritualization (e.g., astragali, aromatic offerings) that cooccurred with evidence for production across contexts that would otherwise be classified differently (e.g., industrial, storage, temple).

The spectrum approach is not limited to specific contextual observations, as we can also identify patterns in ritualization over time. For example, in the Iron Age Southern Levant, we do not see the formal qualities of repetition that define temples of the Middle Bronze Age nor the continuity that marks the Syrian *langraum* tradition from the Bronze through Iron Ages. William Mierse (2012, pp. 300–2) has described this as a shift from prestigious to vernacular idioms in architecture. We think we can be more precise. A spectrum approach suggests that in the Iron Age the focus of ritualization shifts from the built environment towards the portable paraphernalia of offerings (e.g., stands, altars and chalices). Built space is less clearly demarcated as sacred and engagement with the metaphysical is more spatially dispersed. This suggests changes in the institutional and communal setting of religious practice between the Bronze and Iron Ages. We have already rejected the suggestion that these changes can be explained on ethno-religious grounds (e.g., Faust 2010, 2019). However, there are region-wide changes to the political economy and political communities, and an increased prominence in intermediary identities (e.g., family, community, lineage) and 'national' deities, all of which may relate to these changes in religious practice. However, making such causal connections requires us to first recognize what we are dealing with when we consider religion in the Iron Age—not types of buildings or correlates of ritual but lived material contexts. Religion was a constant in the Iron Age, but one made visible strategically, setting some things apart and not others in order to create new relationships between people, spaces and the divine. **Author Contributions:**

Conceptualization, D.J.H.H. and B.R.; methodology, D.J.H.H. and B.R.; validation, D.J.H.H. and B.R.; writing—original draft preparation, D.J.H.H. and B.R.; writing—review and editing, D.J.H.H. and B.R.; visualization, D.J.H.H. and B.R.; supervision, D.J.H.H. and B.R. Both authors have read and agreed to the published version of the manuscript.

**Funding:** This research received no external funding.

**Conflicts of Interest:** The authors declare no conflict of interest.

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
