# Peer review of "Between Rocks and ‘High Places’: On Religious Architecture in the Iron Age Southern Levant"

_religions, doi:10.3390/rel12090740_

Round 1
Reviewer 1 Report
This is an interesting paper about the improvement of the understanding of the relation between religion and architecture using the spectrum approach for the religious architecture in the Iron Age Southern Levant .
By the way, I think that some improvements are necessary to allow a better understanding of the paper by the reader.
First of all, a better and clear hierarchic organization of section 4 "Between rocks and ‘high places’: applying the spectrum approach" is necessary: the forms used to define the architectural tipologies ("Wayside Shrines?", "Industrial cults?") and the case studies (as "Kuntillet ˁAjrud", " Tel Reḥov Apiary", "Khirbat al-Mudayna, Building 200" , "Taanach “Cultic Structure”, "Tell eṣ-Ṣafi Area D Temple") are the same. I think that it would be better to identify the architectural tipologies also with a sub-section number (e.g. "4.1.Wayside Shrines?", "4.2. Industrial cults?") .
I think also that it would be necessary to add figures and planimetries referred to the archaeological study cases used in section 4: it is hard to accept to explain relation between architecture and religion, though through a spectrum approach, without images of the analyzed architectural, material and archaeological contexts.
I suggest to the authors a careful revision of the english editing, e.g.:
- Revised the use of English possessive (it can’t be used if the “owner” is an object):
-
- Line 72: “region’s heterogeneity”
- Line 301: “context’s past”
- Line 453: “Building A’s entrance”
- Line 459: “The site’s association”
- Line 614 “structure’s architecture“
- Line 642 “building’s destruction”
- Line 518 “In seems clear in..”: probably it would be “It seems clear in…”?
Author Response
In response to Reviewer 1 we have :a) included illustrations of our key archaeological examples; b) made the suggested grammatical corrections; c) renumbered our subsections using a 3-tiered system that allows all sections to be labelled consistently throughout the manuscript.
Reviewer 2 Report
The paper deals with a remarkably interesting topic, namely the study on the religious architecture in Iron Age Southern levant regarding a different approach on the concept on which religious behavior can be expressed.
This paper explores a proposal of a different way of approaching evidence for religious practice in the archaeological record, viewing religion as one dimension of social action made visible along a spectrum of ritualization.
The paper title accurately reflects the content and purpose of the paper. The abstract is concise and provides sufficient information. The keywords are adequate. The introduction section presents a relatively good literature review and locates well the work. The article describes very well the methodology and research methods. The check list approach is discussed and presented as a complementary literature review which points directions on the research materialization. It should be underlined the use of the taxonomic approach and its problems. After this it should be highlighted the spectrum approach. The diversity and similarity across categories are taken into consideration,
The results are consistent and represent a contribution to a different way of researching these matters contributing to a different view on the subject which can be further discussed across fields of work. The work is publishable, but it still needs some minor improvements before to be accepted. The conclusion could reflect the discussion and how such discussion actually fits in this SI which can easily be overcome by the authors.
Author Response
In response to Reviewer 2 we have a) revised the conclusion so as to refer back more explicitly to the observations in our paper. We have also reorganised the numbering of our sub-sections into a three-tiered. This addresses comments from both Reviewer 1 & 2.
Reviewer 3 Report
This is an absolutely brilliant article, and I benefitted a great deal from a thorough reading. The publication promises to add a great deal to the discussion of how religious structures and cult activities have been variously identified, and the initial part of the text that focuses on scholarly interpretations and where those interpretations have been found wanting is critical. I have taken the liberty to correct some cosmetic (punctuation and grammatical) errors...but the manuscript is nearly ready for publication.

Author Response
In response to Reviewer 3 a) we have made the grammatical corrections suggested in the annotated manuscript.
Round 2
Reviewer 1 Report
Dear authors,
thank you for the realised modification.